

# Numerical Investigation of Typhoon Waves Generated by Three Typhoons in the China Sea

Qing Shi [1], Jun Tang [1], Yongming Shen [1, 2], Yuxiang Ma [1]

[1] State Key laboratory of Coastal and Offshore Engineering, Dalian University of Technology, Dalian 116023, China
[2] Institute of Environmental and Ecological Engineering, Guangdong University of Technology, Guangzhou 510006, China

*Correspondence to*: Jun Tang (jtang@dlut.edu.cn)

**Abstract.** The typhoon waves generated in the China Sea during the Chan-hom (1509), Linfa (1510) and Nangka (1511) typhoons that occurred in 2015 were numerically investigated. The wave model was based on the a third generation spectral wind-wave model SWAN, in which the wind fields for driving waves were derived from the ERA-interim (ECMWF), CFSv2
(The NCEP Climate Forecast System Version 2) and CCMP (Cross-Calibrated Multi-Platform) datasets. The numerical results were validated using buoy data and satellite observation data. The simulation results under the three types of wind fields were in good agreement with the observed data. The CCMP wind data was the best in simulating waves overall, and the wind speeds pertaining to ERA and CCMP were notably smaller than those observed near the typhoon centre. The Holland wind model was used to revise and optimize the wind speed pertaining to the CCMP near the typhoon centre, and the wind speed correction
coefficient, correction formula and corresponding parameters were determined. Based on these findings, the CCMP and CCMP/Holland blended wind fields were used to simulate the typhoon waves generated during the Meranti (1614), Rai (1615) and Malakas (1616) typhoons that occurred in September 2016. A comparison between the simulated wave heights and those obtained from the Jason-2 altimeter data indicated that all correlation coefficients between the simulated values and the satellite observations were greater than 0.75. The blended wind field was better overall in simulating the wave heights. The simulated
maximum wave heights were more similar to the satellite observations, and the root mean square error of the blended wind field was 0.223 m lower than that of the CCMP. The results demonstrated that the CCMP wind-driven SWAN model could appropriately simulate the typhoon waves generated by three typhoons in China Sea, and the use of the CCMP/Holland blended wind field could effectively improve the accuracy of typhoon wave simulations.

## 1 Introduction

The mechanism of development of typhoon waves caused by three typhoons in China's offshore region is complicated. Three typhoons co-exist and affect offshore zones in China almost every year, and the development and evolution of three simultaneous typhoons are more complex and difficult to predict than those of a single typhoon. Multiple typhoons inevitably interact with each other, and their influence on waves is considerably different from that of a single typhoon. As the main input energy of ocean wave movement, different wind fields lead to different wave distribution characteristics and evolution laws.



The swells generated by multiple typhoons, as well as the typhoon waves and swell waves in adjacent waters, tend to form a complex mixed wave, which can cause considerable damage to coastal engineering, ship navigation and marine engineering applications. Therefore, it is crucial to study the complicated typhoon waves caused by three simultaneous typhoons.

Several studies focused on numerical simulations of typhoon waves have been conducted. Zhang et al. (2011) analysed the reliability of the CCMP wind field and simulated the wave process in the Bohai Sea by using the SWAN model, in which the
CCMP wind field was used as the forced input. Kuang et al. (2015) compared the plane distribution and time variation characteristics of three types of sea surface wind fields (CCMP, NCEP, and ERA) in the Taiwan Strait wind field and analysed and evaluated the errors of the three wind fields by using the wind speed and wind direction data for 2011, which were obtained using observations from buoys. Abdalla et al. (2010) and Queffeulou et al. (2011) verified the wind speed and significant wave height obtained using Jason-2 by comparing the data with those obtained from buoys. Stopa et al. (2014) compared the wind
speeds and wave heights from ERA-I and CFSR and utilized the same set of altimetry and buoy observations and error metrics to assess the consistency of the data in time and space. Wang et al. (2016) conducted a comprehensive analysis of the accuracy of significant wave height obtained from Jason-2 based on the long-term observations (from 2008 to 2014) of hydrometeorological buoys in the Bohai sea, the Yellow Sea, the East China Sea and the South China Sea. Zhou et al. (2016) used the NCEP-FNL reanalysis wind field and WW3 to simulate the simultaneous occurrence of typhoons 1509 Can-hom,
1510 Linfa and 1511 Nangka in 2015. Pan et al. (2016) focused on the improvement of wind field hindcasts for two typical tropical cyclones, i.e., Fanapi and Meranti, which occurred in 2010. Liang et al. (2016) investigated the wave climate of the Bohai Sea, Yellow Sea, and East China Sea for the period from 1990 to 2011 by using the SWAN. Shao et al. (2018) analysed 29 tropical cyclones (TCs) in the South China Sea (SCS) and East China Sea (ECS) for a period of four years (2011–2014) at 10 buoy locations. It was found that the ERA-Interim largely under predicted the wind speeds near the TC centre, although the
Holland model performed generally satisfactorily for that location. A formula for blended TC wind fields combining two datasets was proposed, which demonstrated the satisfactory capacity of the TC wind simulation. Next, the blended wind model was applied in TC wave simulations in the SCS and ECS, and it was noted to demonstrate better performance than that of both the ERA-interim and the Holland model. Jiang et al. (2018) utilized ERA-interim reanalysis data modified using a parametric typhoon model to simulate the waves and surges over the northwest pacific region for a 35-year period. Wang et al. (2018)
investigated the extreme wave climate variability in the South China Sea (SCS) using the significant wave height (SWH) data simulated by the third-generation wave model WAVEWATCH-III (WW-III) for the period 1976–2014.

The abovementioned studies improved our insight into the influence of wind fields on the accuracy of typhoon wave simulations. However, the existing studies focused primarily on typhoon waves generated by a single typhoon, and the simulation area was usually small. It was thus difficult to observe the distribution characteristics and evolution law of wind
waves in different sea areas. Moreover, compared to the progress of numerical simulations of typhoon waves caused by a single typhoon, the simulations of typhoon waves in a unique weather background involving three typhoons has not been extensively investigated, and there is a lack of research on the wind fields that exist during the occurrence of three typhoons. In the wind¬–wave model, wind plays a key role in the wave simulations, and establishing an accurate wind field is the basis





for achieving an accurate simulation of the waves. At present, a variety of wind fields are commonly used worldwide. However,
only a small number of studies have attempted to compare and evaluate these wind fields in the simulation of typhoon waves
in China Sea.

The present study intended to perform a comparative analysis of the typhoon waves in China Sea generated under the influence
of three typhoons by using the ERA-interim, CFSv2 and CCMP wind-driven SWAN models. Subsequently, the wind field
leading to the best performance of the wave simulation was studied and revised, and the temporal and spatial distribution
characteristics and simulation accuracy of the wave field before and after the correction of the wind field were compared and
analysed. The remaining paper is organized as follows: Section 2 presents the principle of the wave model, modelling data and
model setup used in the study. Section 3 presents the validation results pertaining to the occurrence of three typhoons in China
Sea in 2015 and 2016 against altimetry and buoy data. In addition, the investigation concerning the improvement of the wind
field, and the comparison of the accuracy of the wave field simulation before and after wind field correction is discussed. The
study's conclusions are presented in Section 4.

## 2 Numerical model and setup

### 2.1 Wave model

SWAN (Booij et al.1996) is a third-generation wave model developed by Delft University of Technology. The model computes
random, short-crested wind-generated waves in coastal regions and inland waters. Further, the model is driven by wind and
can simulate wave diffraction, refraction, wave breaking, and wave increase and decrease. The control equation for the SWAN
wave model is the spectral action balance equation, which can be expressed in Cartesian coordinates as

$$\frac{\partial N}{\partial t} + \frac{\partial C_x N}{\partial x} + \frac{\partial C_y N}{\partial y} + \frac{\partial C_\sigma N}{\partial \sigma} + \frac{\partial C_\theta N}{\partial \theta} = \frac{S_{tot}}{\sigma} \tag{1}$$

$$N(\sigma, \theta) = \frac{E(\sigma, \theta)}{\sigma} \tag{2}$$

where $N(\sigma, \theta)$ is the action density, $E(\sigma, \theta)$ is the energy density, $\sigma$ represents the frequency, $\theta$ is the wave direction, $C_x$ and
$C_y$ are the propagation velocities of the wave energy in spatial space, and $C_\sigma$ and $C_\theta$ are the propagation velocities in the
spectral space caused respectively by the change in current and water depth. These parameters can be expressed as in the
following expressions:

$$\begin{pmatrix} C_x \\ C_y \end{pmatrix} = \begin{pmatrix} \frac{1}{2}\left(1 + \frac{2|\vec{k}|h}{sinh(2|\vec{k}|h)}\right)\frac{\sigma\vec{k}}{|\vec{k}|^2} \cdot \vec{\imath} + \vec{U}.\vec{\imath} \\ \frac{1}{2}\left(1 + \frac{2|\vec{k}|h}{sinh(2|\vec{k}|h)}\right)\frac{\sigma\vec{k}}{|\vec{k}|^2} \cdot \vec{\jmath} + \vec{U}.\vec{\jmath} \end{pmatrix} \tag{3}$$





$$C_\sigma = \frac{\partial \sigma}{\partial h}\left(\frac{\partial h}{\partial t} + \vec{U} \cdot \nabla_{\vec{x}} h\right) - C_g \vec{k} \cdot \frac{\partial \vec{U}}{\partial s} \tag{4}$$

$$C_\theta = -\frac{1}{|\vec{k}|}\left(\frac{\partial \sigma}{\partial h}\frac{\partial h}{\partial m} + \vec{k} \cdot \frac{\partial \vec{U}}{\partial m}\right) \tag{5}$$

$$\nabla_{\vec{x}} = \left(\frac{\partial}{\partial x}, \frac{\partial}{\partial y}\right) \tag{6}$$

where $\overrightarrow{C_g}$ is the wave group velocity, $\overrightarrow{U}$ is the current velocity, $\vec{k}$ is the unit vector, $h$ is the water depth, $s$ is the space coordinate along the $\theta$ direction, and $m$ is the space coordinate perpendicular to $\theta$.

$S_{tot}$ is the source term that represents all physical processes which generate, including the wind energy input, whitecapping, depth-induced wave breaking, non-linear wave–wave interaction, bottom friction dissipation, vegetation dissipation, and sediment dissipation. The source term can be expressed as

$$S_{tot} = S_{in} + S_{ds,w} + S_{ds,b} + S_{ds,br} + S_{nl4} + S_{nl3} + S_{ds,veg} + S_{ds,mud} \tag{7}$$

where $S_{in}$ represents the wind energy input (Phillips, 1957, Miles, 1957), $S_{ds,w}$ represents whitecapping, $S_{ds,b}$ is the bottom friction dissipation term, $S_{ds,br}$ represents depth-induced wave breaking, $S_{nl4}$ represents the quadruplet wave–wave interactions, $S_{nl3}$ represents the triad wave interaction, $S_{ds,veg}$ is the vegetation dissipation term, and $S_{ds,mud}$ is the sediment dissipation term. The present study focused mainly on typhoon waves, and the wind energy input, bottom friction dissipation, quadruplet wave–wave interaction, whitecapping and depth-induced wave breaking were primarily considered, among which the wind energy input term was the main source term.

The wind energy input item $S_{in}$ can be expressed as

$$S_{in}(\sigma, \theta) = A + BE(\sigma, \theta) \tag{8}$$

where $A$ represents linear growth and $B$ represents exponential growth. The expression for the term $A$ was obtained from Cavaleri and Malanotte-Rizooli (1981):

$$A = \frac{1.5 \times 10^{-3}}{2\pi g^2}\left(U_* \max\left(0, \cos(\theta - \theta_w)\right)\right)^4 H \tag{9}$$

$$H = \exp\left(-\left(\frac{\sigma}{\sigma_{PM}^*}\right)^{-4}\right) \tag{10}$$

$$\sigma_{PM}^* = \frac{0.13g}{28U_*}2\pi \tag{11}$$

where $U_*$ is the frictional velocity, $\theta_w$ is the wind direction, $H$ is the filter used to avoid exponential growth of low-frequency waves, and $\sigma_{PM}^*$ (Pierson et al., 1964) is the peak frequency of the fully developed sea state. The expression of B was obtained from Komen et al. (2009):





$$B = \max\left(0, 0.25 \frac{\rho_\mathrm{a}}{\rho_\mathrm{w}}\left(28 \frac{U_*}{c_\mathrm{ph}}\cos(\theta - \theta_\mathrm{w}) - 1\right)\right)\sigma \qquad (12)$$

in which $c_\mathrm{ph}$ is the phase speed; and $\rho_\mathrm{a}$ and $\rho_\mathrm{w}$ denote the densities of air and water, respectively.

## 2.2 Introduction of data

ETOPO1 global topographic and bathymetric data (Amante et al., 2009) with a resolution of 1′ can satisfy the requirements of global sea wave simulations.

ERA-interim is the wind reanalysis dataset, which represents an improved atmospheric model and assimilation system

compared to its predecessor ERA-40 (Dee et al., 2011) of the European Centre for Medium-Term Weather Forecasting (ECMWF). ERA-interim utilizes the 4D-VAR (4-Dimensional Variational Data assimilation) scheme used in reanalysis. This system contains a coupled wave–atmosphere component, and the wave model assimilates the altimeter observations (Stopa, J.E., 2018).

The CFSv2 was executed in 2011, and it has demonstrated improvement in the product, especially in the tropical regions, with

an increased resolution of 22 km (T574) (Saha et al., 2014). CFSv2 uses 3D-VAR (3-Dimensional Variational Data assimilation) with assimilations being updated every 6 h (Stopa, J.E., 2018).

The CCMP dataset (Atlas et al., 2011) is provided by the PO.DAAC (Physical Oceanographic Data Distribution Archives Center) of NASA (National Aeronautics and Space Administration), and it uses the reanalysis and operational data of ECMWF as the background field. Variational assimilation analysis (VAM) is used, which combines SSM/I, TMI, AMSR-E, QuikSCAT,

ADEOS-II and other satellite wind data, as well as ship and buoy observation data (Atlas et al., 1996). Studies have shown that CCMP has a considerably higher accuracy than wind field data measured by other single satellite platforms (Atlas et al., 2008).

Jason-2 is a satellite that was designed specifically for global ocean observation, including high-precision sea level detection and meteorological observation; it was launched by NASA and EUMETSAT (European Meteorological Satellite Organisation)

on June 20, 2008. Jason-2 provides reliable and detailed oceanographic data. The data from Jason-2's geographical data record (GDR), which has been completely corrected, was selected for use in the present study. The data accuracy is high owing to the high orbit accuracy of GDR and the waveform being re-corrected. The significant wave height products observed by Jason-2 include those pertaining to the Ku-band and C-band. In the product specification of OSTM/Jason-2 in 2011, Ku-band observations are better than C-band. Therefore, the study used Ku band data with a data accuracy of 0.001 m (Wang et al.

140    2016).

## 2.3 Topography and model setup

The range of calculation was 105˚E–145˚E, 0˚N–40˚N, which covers the China sea area. The tracks of the three typhoons that occurred in 2015 are shown in Fig. 1. Four buoys were placed in the Yellow Sea, East China Sea and South China Sea, and





the topography and buoy positions in the calculation area are shown in Figure 2. The buoy data is referenced from Zhou et al.

145   (2016).

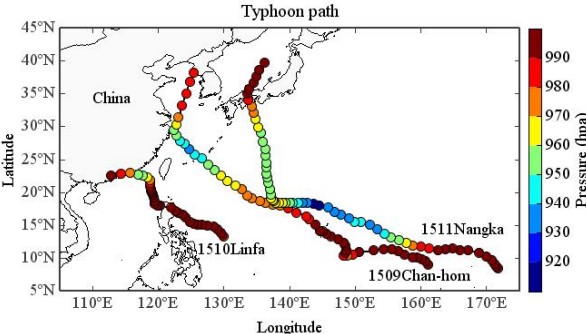

Fig. 1. Tracks of three typhoons in 2015

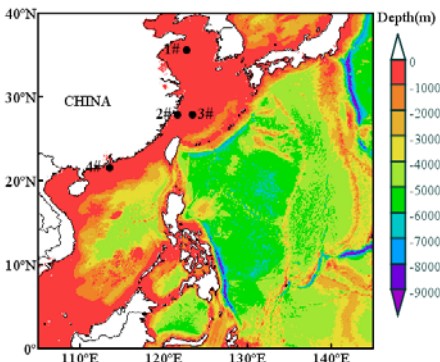

Fig. 2. Topographic and position of the buoy

The resolution of the computing grid in space was 0.1˚. The range of direction angle was 0–360˚, which was divided into 36
grids. The range of frequency was 0.04–1 Hz. In the calculation, the wind input, non-linear interaction, bottom friction
dissipation, depth-induced wave breaking and wave diffraction were taken into account. First, the SWAN model was driven
via ERA, CFSv2 and CCMP wind fields. The period of calculation was from 0:00 on July 1, 2015 to 18:00 on July 18 (UTC).
Because the buoys were close to the coast, the self-nesting mode of SWAN was used to verify the buoy data (no nesting is

required when verifying satellite data). The accuracy of the nested computing grid in space was 0.02˚. The depth of the
topography and position of the buoy in the nested area are shown in Figure 3.





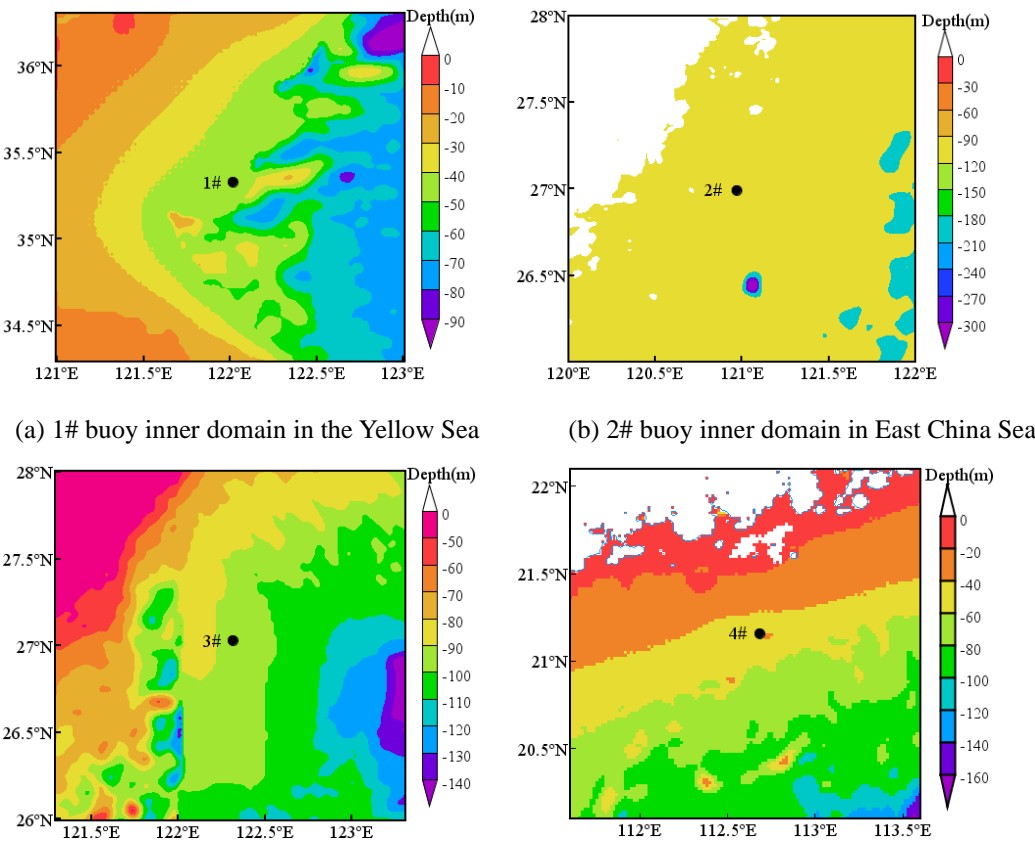

(a) 1# buoy inner domain in the Yellow Sea    (b) 2# buoy inner domain in East China Sea

(c) 3# buoy inner domain in the East China Sea    (d) 4# buoy inner domain in the South China Sea

Fig. 3. The topographic and position of the buoy in nested calculation area

## 3 Simulation of typhoon waves during three typhoons

### 3.1 Simulation of typhoon waves during three typhoons that occurred in 2015

The Pearson correlation coefficient, mean deviation (Bias) and root mean square error (RMSE) were used for data deviation analysis. The parameters of the evaluation used for verification included the significant wave height obtained from buoy observation and the significant wave height and wind speed obtained from Jason-2 observation. Here, $X_{SL}$ represents the calculated results and $X_{MEA}$ denotes the buoy altimetry or satellite data. The relevant expressions are as follows:

$$\text{Bias} = \frac{1}{n}\sum_{1}^{n}(X_{SL} - X_{MEA}) \tag{13}$$

$$\text{RMSE} = \sqrt{\frac{1}{n}\sum_{1}^{n}(X_{SL} - X_{MEA})^2} \tag{14}$$





The simulated wave height at different buoys during the three typhoons were compared with the observed data, and the results are shown in Fig. 4 and Table 1. It can be seen that the correlation coefficients of the simulation results of two buoys driven

by the three wind fields in the East China Sea are as much as 0.966, and the correlation coefficient of buoy #1 in the Yellow Sea is also more than 0.8. The calculated wave height at buoy #4 in the South China Sea is similar to that observed, but the correlation coefficient is relatively low compared with those at other buoys. This finding occurs because the Yellow Sea and East China Sea are open sea areas, for this reason, the wave propagation is less affected by topography. However, the topography of the South China Sea is complicated, as islands and reefs are densely distributed in the sea area, and waves

undergo complex physical processes, such as reflection, refraction, diffraction and fragmentation. In addition, the buoy points of the South China Sea are close to the coast, and an orthogonal curve grid is used to describe the topography of the shallow coastal areas. All of these factors lead to the correlation coefficient of the buoy points in the South China Sea being lower than the correlation coefficients for buoys in the Yellow Sea and East China Sea. Generally, for a small wave height, the simulation results of CCMP are more similar to the measured values, and the results of ERA and CFSv2 are smaller. For a large wave

height, the CFSv2 wind field demonstrates the best performance, while the results of ERA and CCMP are smaller. According to the interaction between wind and waves, gales usually generate large waves. It can be inferred that the wind speeds near the typhoon centre in the study area, pertaining to the ERA and CCMP, are smaller than the actual values. The wind field of CCMP is more reasonable when the wind speed is low, this aspect is discussed further in the subsequent sections.

In combination with the analysis presented in Table 1, the absolute values of the Bias and RMSE of buoy #2 are determined

to be less than 0.2 m, which represents the best performance. The Bias and RMSE of buoy #4 are small. Buoy #3 presents remarkably satisfactory trends, with the maximum Bias and RMSE being -0.796 m and 1.038 m, owing to the large wave height. The overall simulation results for buoy #1 can be considered moderate, as determined by the morphological characteristics of the original wind field.

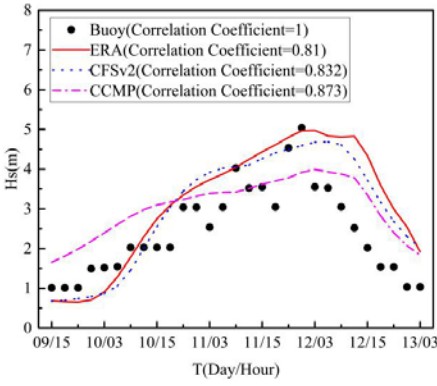

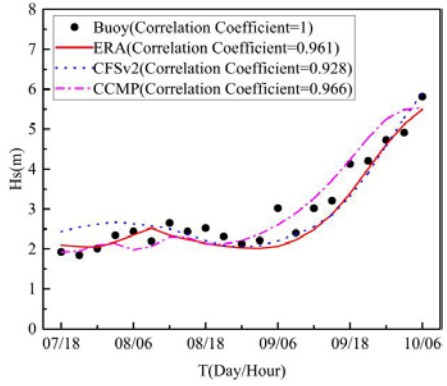

(a) Wave height of buoy 1# in the Yellow Sea    (b) Wave height of buoy 2# in the East China Sea




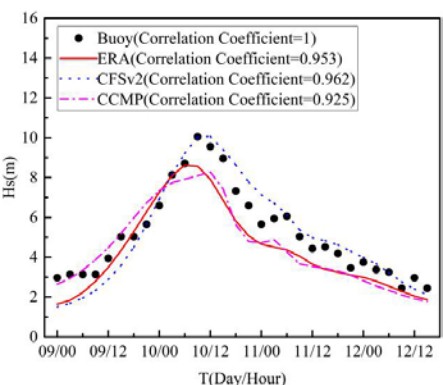

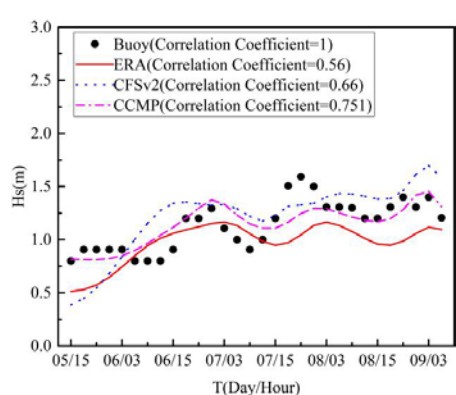


(c) Wave height of buoy 3# in the East China Sea

(d) Wave height of buoy 4# in the South China Sea

Fig. 4 Verification of wave height of buoys

Table 1 Analysis of wave height of buoy

| Buoy | Deviation of Wave Height | ERA | CFSv2 | CCMP |
|---|---|---|---|---|
| 1# | CC | 0.81 | 0.832 | 0.873 |
| | Bias (m) | 0.703 | 0.558 | 0.592 |
| | RMSE (m) | 1.121 | 0.965 | 0.84 |
| 2# | CC | 0.81 | 0.832 | 0.873 |
| | Bias (m) | -0.196 | -0.034 | 0.041 |
| | RMSE (m) | 0.129 | 0.17 | 0.116 |
| 3# | CC | 0.953 | 0.962 | 0.925 |
| | Bias (m) | -0.796 | 0.035 | -0.623 |
| | RMSE (m) | 1.037 | 0.788 | 1.038 |
| 4# | CC | 0.56 | 0.66 | 0.751 |
| | Bias (m) | -0.165 | 0.085 | 0.008 |
| | RMSE (m) | 0.261 | 0.261 | 0.154 |

To further verify the model, the altimetry and sea surface wind speed obtained from Jason-2 as it passed through the simulated sea during the occurrence of typhoon waves were adopted. The satellite trajectory points were divided into three categories: those pertaining to the Yellow Sea (119˚E–127˚E, 31˚N–39˚N), the East China Sea (117.18˚E–131˚E, 23˚N–31.18˚N) and the South China Sea (105˚E–119˚E, 4˚N–23˚N). The detailed verification results are shown in Fig. 5.




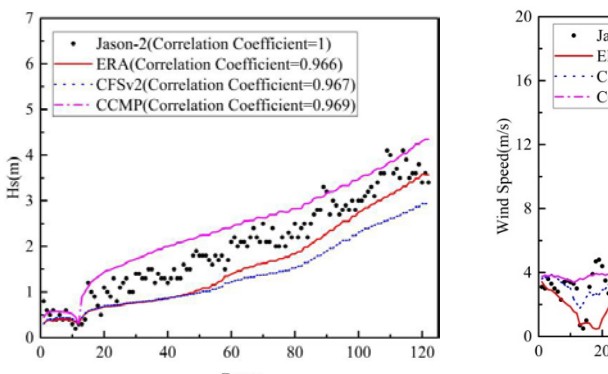

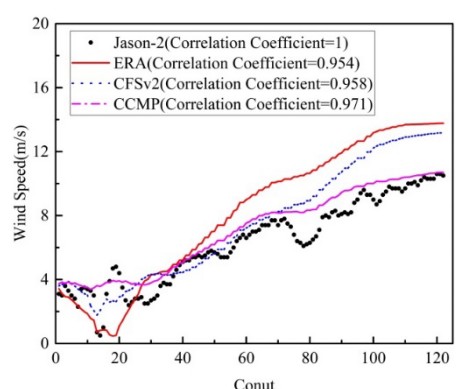

(a1) Wave height verification in the Yellow Sea

(a2) Wind speed verification in the Yellow Sea

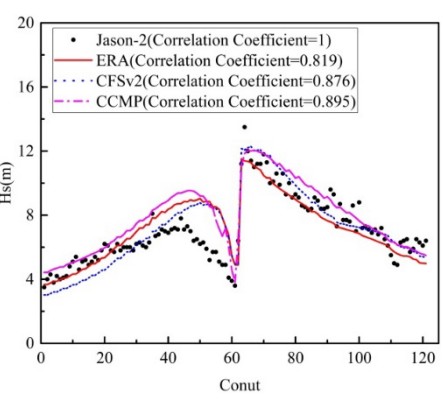

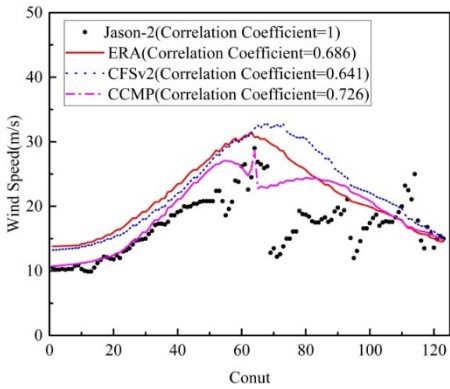


(b1) Wave height verification in the East China Sea

(b2) Wind speed verification in the East China Sea

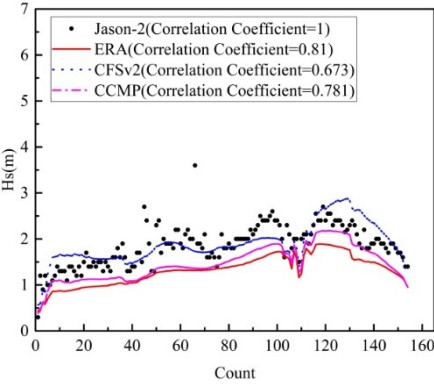

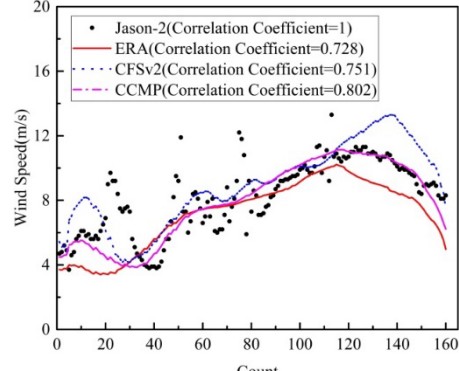

(c1) Wave height verification in the South China Sea

(c2) Wind speed verification in the South China Sea

Fig. 5 validation of wave height and wind speed of satellite



The comparison results of Jason-2 presented in Table 2 show that the correlation coefficients of the simulated wave height and wind speed between the satellite observations are relatively high under the three wind fields. The correlation coefficients of

the wave height and wind speed reach up to 0.969 and 0.971. Even at the South China Sea, the correlation coefficients are more than 0.7. Overall, the trend of the wave height is similar to that of the wind speed. The results show that the RMSEs of the wave height in the Yellow Sea and South China Sea are smaller than 1 m. Because of the greater wave height, the maximum wave height in the East China Sea is as much as 12 m, and the RMSE of the wave height is larger, although the curve of the wave height in the East China Sea fits well with the observed data. The wind field comparison results indicate that the CCMP

wind field fits the satellite observation better, followed by the fitting of the CFSv2 wind field. The CFSv2 wind field data is slightly larger, and the CCMP wind field and CFSv2 wind field are more similar to the satellite data than the ERA wind field is.

Table 2 Data Comparison of Jason-2

| Sea | Deviation of Wave Height | ERA | CFSv2 | CCMP |
|---|---|---|---|---|
| The Yellow Sea | CC | 0.966 | 0.967 | 0.969 |
| | Bias (m) | -0.415 | -0.483 | 0.423 |
| | RMSE (m) | 0.491 | 0.714 | 0.495 |
| The East China Sea | CC | 0.819 | 0.876 | 0.895 |
| | Bias (m) | 0.148 | 0.01 | 0.838 |
| | RMSE (m) | 1.232 | 1.176 | 1.281 |
| The South China Sea | CC | 0.81 | 0.673 | 0.781 |
| | Bias (m) | -0.42 | 0.005 | -0.413 |
| | RMSE (m) | 0.618 | 0.337 | 0.509 |
| Sea | Deviation of Wind speed | ERA | CFSv2 | CCMP |
| The Yellow Sea | CC | 0.954 | 0.958 | 0.971 |
| | Bias (m/s) | 1.872 | 1.236 | 0.773 |
| | RMSE (m/s) | 2.7 | 1.8 | 0.995 |
| The East China Sea | CC | 0.686 | 0.641 | 0.726 |
| | Bias (m/s) | 4.4 | 5.4 | 2.4 |
| | RMSE (m/s) | 6 | 7.1 | 4.4 |
| The South China Sea | CC | 0.728 | 0.751 | 0.802 |
| | Bias (m/s) | -0.536 | 0.04 | -0.389 |
| | RMSE (m/s) | 0.597 | 0.355 | 0.479 |





To study the typhoon waves corresponding to three typhoons accurately, it is necessary to verify the accuracy of the description of the typhoon centres. The maximum average wind speed near the centre of the three typhoons (the maximum average wind speed pertaining to the three wind fields at the corresponding time) and the BEST TRACK values (tcdata.typhoon.org.cn) (2014) were compared, and the results are shown in Fig. 6.

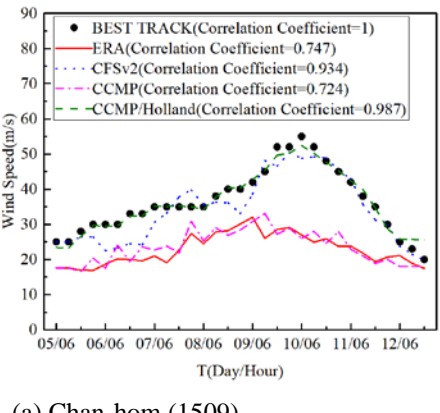

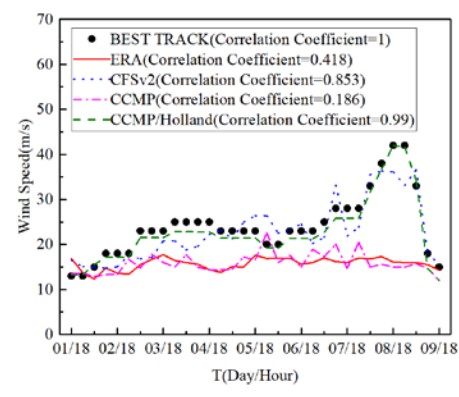

(a) Chan-hom (1509)

(b) Linfa (1510)

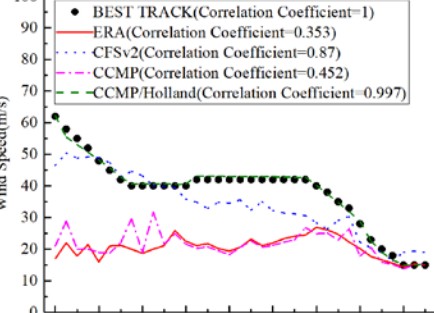


(c) Nangka (1511)

Fig. 6 Comparison of maximum wind speed near the center of typhoon

It can be seen from Fig. 6 that the values of the maximum wind speed pertaining to the CFSv2 wind field, for which the correlation coefficients are more than 0.85, is the most similar to the BEST TRACK values, whereas the wind speeds pertaining to the ERA and CCMP wind fields are notably smaller in comparison.

To accurately simulate the waves during three typhoons, the CCMP wind field, which demonstrated the best performance in terms of the wave height and wind speed in the simulation of typhoon waves in three typhoons that occurred in 2015, was selected as the background wind field, and it was blended with the Holland model (1980) to overcome the shortcomings of CCMP, these shortcomings include the wind speed near the typhoon centre being small and the deviation of the typhoon centre.





The Holland model is based on the exponential distribution of the atmospheric pressure field defined by Schloemer (1954) and

a peak parameter B (1980), and it can be expressed as follows:

$$P = P_c + (P_n - P_c)\exp\left(-\frac{R_{max}^B}{r^B}\right) \tag{15}$$

$$U_g = \sqrt{\frac{B}{\rho_a}\left(\frac{R_{max}}{r}\right)^B (P_n - P_c)exp\left[-\left(\frac{R_{max}}{r}\right)^B\right] + \left(\frac{rf^2}{2}\right)} - \frac{rf}{2} \tag{16}$$

$$R_{max} = 28.52\tanh[0.0873(\varphi - 28)] + 12.22\exp\left(\frac{P_c - P_n}{33.86}\right) + 0.2V_f + 37.2 \tag{17}$$

$$B = 1.5 + \frac{980 - P_c}{120} \tag{18}$$

where $U_g$ represents the gradient wind speed at radius $r$, $f$ is the Coriolis force parameter, $\rho_a$ is the air density, $P_n$ is the atmospheric pressure around the typhoon, $P_c$ is the atmospheric pressure at the typhoon centre, $R_{max}$ is the maximum wind speed radius and B is the parameter defined by Hubert (1991).

The combined scheme can be defined with reference to Wang et al. (2017). First, according to the position of the typhoon centre, the radius of the circle that pertains to the most similar wind speed of the theoretical wind field and that of the CCMP

wind field was determined as the reference superposition radius $R$. $R1$ and $R2$ denote the characteristic superposition radius. $R1$ is determined from the inward $L1$ distance and $R2$ is determined from the outward $L2$ distance based on $R$. To ensure smooth integration of the theoretical wind field and CCMP reanalysis of the wind field, $L1$ and $L2$ are generally defined to be between $0.05\,R$ and $0.15\,R$. The superposition formula can be defined as follows:

$$\begin{cases} V_x = V_{Hx} \ , \\ V_y = V_{Hy} \ , \end{cases} \quad r < R1 \tag{19}$$

$$\begin{cases} V_x = \alpha V_{CCMPx} + (1-\alpha)V_{Hx} \ , \\ V_y = \alpha V_{CCMPy} + (1-\alpha)V_{Hy} \ , \end{cases} \quad R1 \leq r < R2 \tag{20}$$

$$\begin{cases} V_x = V_{CCMPx} \ , \\ V_y = V_{CCMPy} \ , \end{cases} \quad r \geq R2 \tag{21}$$

$$\alpha = \frac{r - R + L1}{L1 + L2} \tag{22}$$

where $V_x$ and $V_y$ represent the components of the superimposed wind speed in the $x$ and $y$ directions, respectively. $V_{Hx}$ and $V_{Hy}$ represent the components of theoretical wind speed in the $x$ and $y$ directions, respectively; and $V_{CCMPx}$ and $V_{CCMPy}$

represent the components of the CCMP wind speed in the $x$ and $y$ directions, respectively.



To adjust the wind speed to the standard 10-m elevation above the sea surface, multiplication by a correction factor K should be performed. Powell (1987) suggested K to be 0.80, Powell and Black (1990) suggested K to be between 0.75–0.80 and Harper and Holland (1999) suggested K to be 0.70. In the present study, the K values for Can-hom, Lifa and Nangka were determined to be 0.82, 0.78 and 0.78, respectively. The superimposed radius R was 2.3 times the maximum typhoon radius.

Next, a blended wind field was used to drive the SWAN model, and the comparison between the simulated wave heights driven by the blended wind field and those driven by the CCMP is shown in Figure 7.

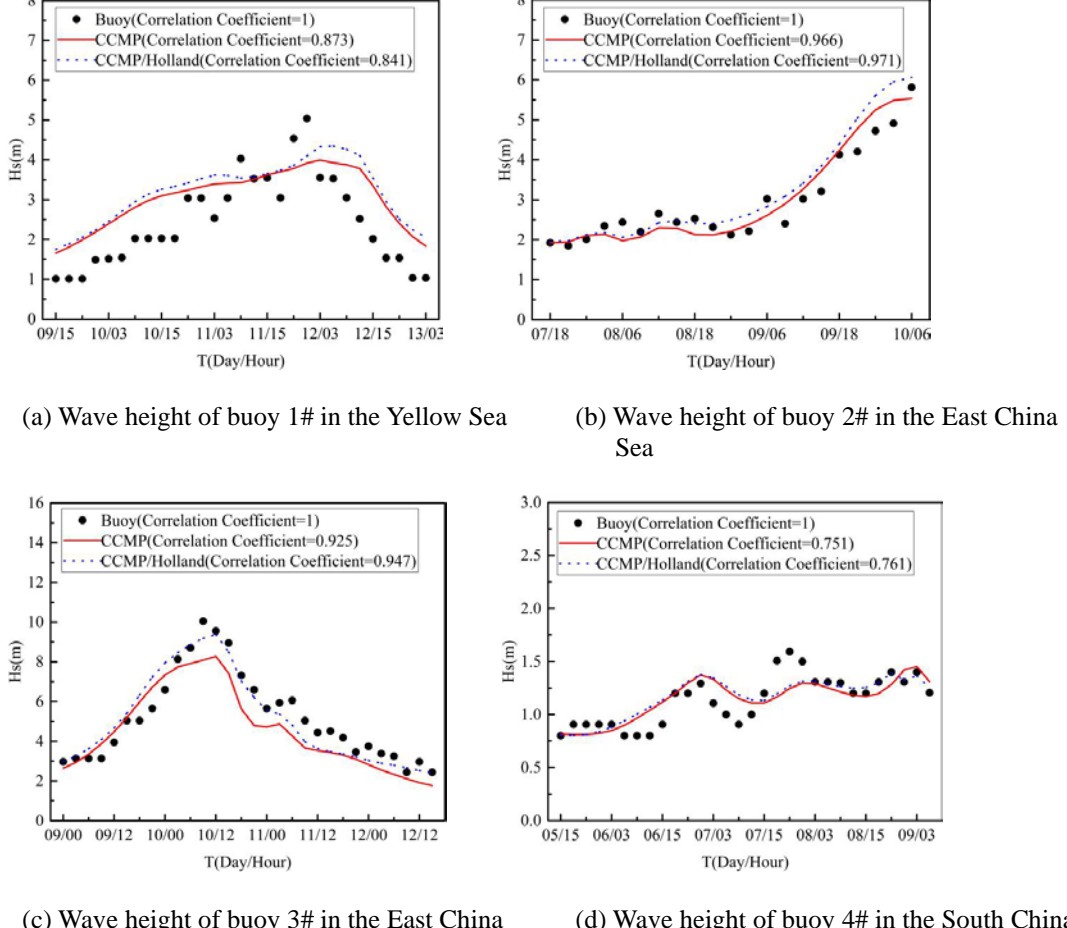

(a) Wave height of buoy 1# in the Yellow Sea

(b) Wave height of buoy 2# in the East China Sea

(c) Wave height of buoy 3# in the East China Sea

(d) Wave height of buoy 4# in the South China Sea

Fig. 7 Comparison of simulated wave height of buoys before and after correction of wind


From Figure 7, it can be seen that the simulated wave height driven by the blended wind field is mostly consistent with the buoy data, and the simulated maximum wave height due to the blended wind field is more consistent with the buoy data. It can be concluded that the blended wind field overcomes a limitation of the CCMP which the wind speed near the centre of typhoon is too small, and the simulated wave driven by the blended wind is more accurate.





In conclusion, the above mentioned verification results obtained using observation data from buoys and satellites indicate that the present model is suitable for modelling waves caused by three typhoons in the China Sea. It is found that the simulated wave height driven by the CCMP wind field is more similar to buoy data overall, and the RMSE is smaller in this case. When the wave height is small, the simulated wave heights driven by the CCMP wind field are similar to the measured data, and the simulated wave heights driven by both ERA and CFSv2 are smaller than the measured data; when the wave height is large,

the simulated wave heights driven by the CFSv2 wind field agree best with the measured data, and those driven by the ERA and CCMP wind fields are smaller than the measured data. It can be concluded that the wind speed of the CCMP wind field far from the centre of typhoon is more reasonable; in addition, this observation is consistent with the results from Kuang et al. (2015). It is further noted that the maximum wind speed near the typhoon centre of the ERA and CCMP wind fields is significantly smaller than the corresponding BEST TRACK value. For the wind speed near the centre of the typhoon, the

blended wind field constructed by using the CCMP-superimposed Holland model is more reasonable. The simulated maximum wave height driven by the blended wind field is more reasonable.

### 3.2 Simulation of typhoon waves due to three typhoons that occurred in 2016

In 2016, three typhoons occurred from September 9th to 20th: Typhoons Meranti 1614 and Malakas 1616 were generated in the Northwest Pacific Ocean, and they invaded the East China Sea and the Yellow and Bohai Seas simultaneously. During this

period, typhoon Rai 1615 also formed in the South China Sea. Typhoon Rai 1615 was not intense and its duration was short; however, it had a considerable impact on the South China Sea. The typhoon tracks are shown in Fig. 8.

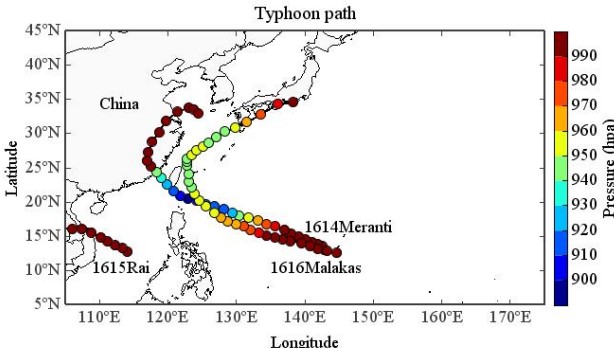

Fig. 8 Tracks of three typhoons in 2015

The differences in the maximum wind speed near the typhoon centre between the different wind fields and the BEST TRACK

data are compared and shown in Fig. 9. The maximum wind speeds pertaining to the three types of wind fields are smaller than the BEST TRACK data, and the difference is more pronounced for the ERA and CCMP wind fields. Based on the previous study results, the CCMP wind field was revised as per the following considerations. Typhoon Rai 1615 had a short duration, and the CCMP wind speed of typhoon Rai was close to the BEST TRACK wind speed; thus, no revision was required in this





case. The correction factor K for typhoons Meranti and Malakas was defined as 0.8. The corrected results are shown in Figure

290    9.

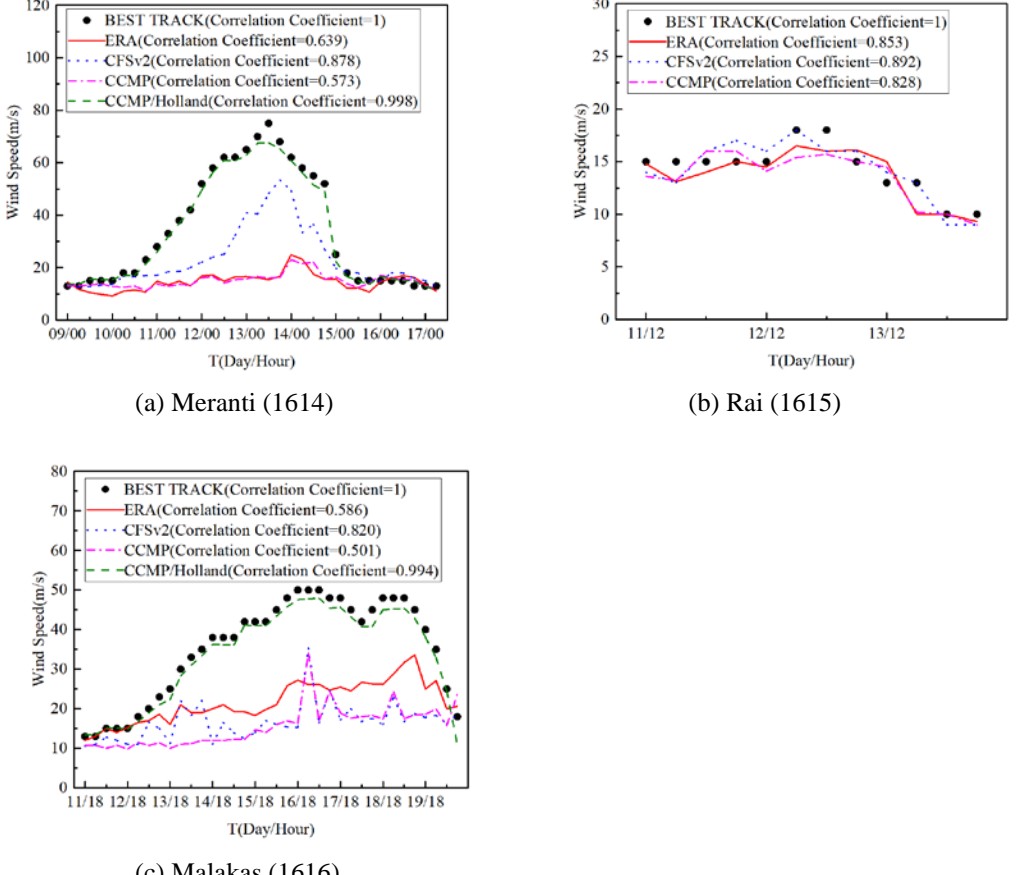

(a) Meranti (1614)                                    (b) Rai (1615)

(c) Malakas (1616)

Fig. 9 Comparison of maximum wind speed for different wind fields near the center of typhoon

The typhoon waves caused by typhoons Meranti, Rai and Malakas that occurred in September 2016 in the China Sea were

simulated by using the SWAN driven by the CCMP and CCMP/Holland blended wind fields. The computational domain and

parameter settings of the model were consistent with those for the typhoons that occurred in 2015, as described in Section 3.1.

The simulation period was from 0:00 on September 7 to 18:00 on September 20, 2016 (UTC). The simulated wave heights

from September 12 to 19 were compared with the wave heights observed using the Jason-2 satellite, as shown in Figure 11.

The selected trajectory points of Jason-2 are shown in Figure 10.


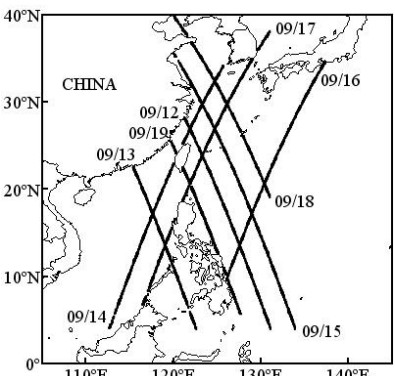

Fig. 10 Satellite trajectory points of Jason-2

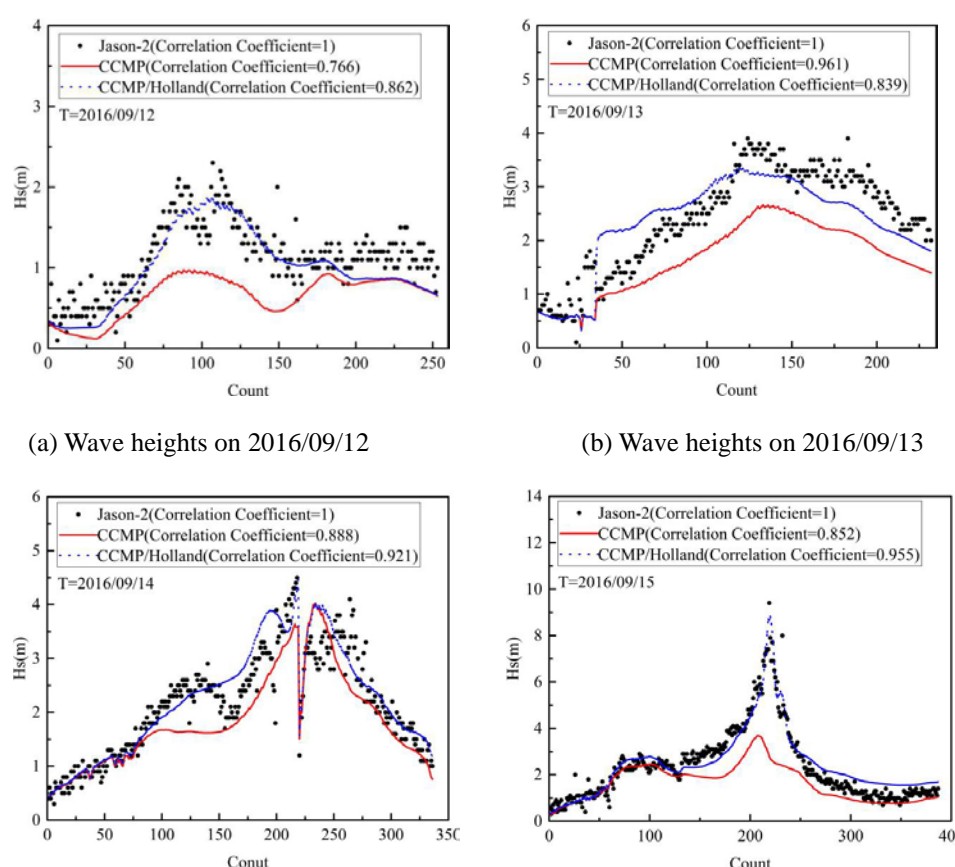

(a) Wave heights on 2016/09/12

(b) Wave heights on 2016/09/13

(c) Wave heights on 2016/09/14

(d) Wave heights on 2016/09/15





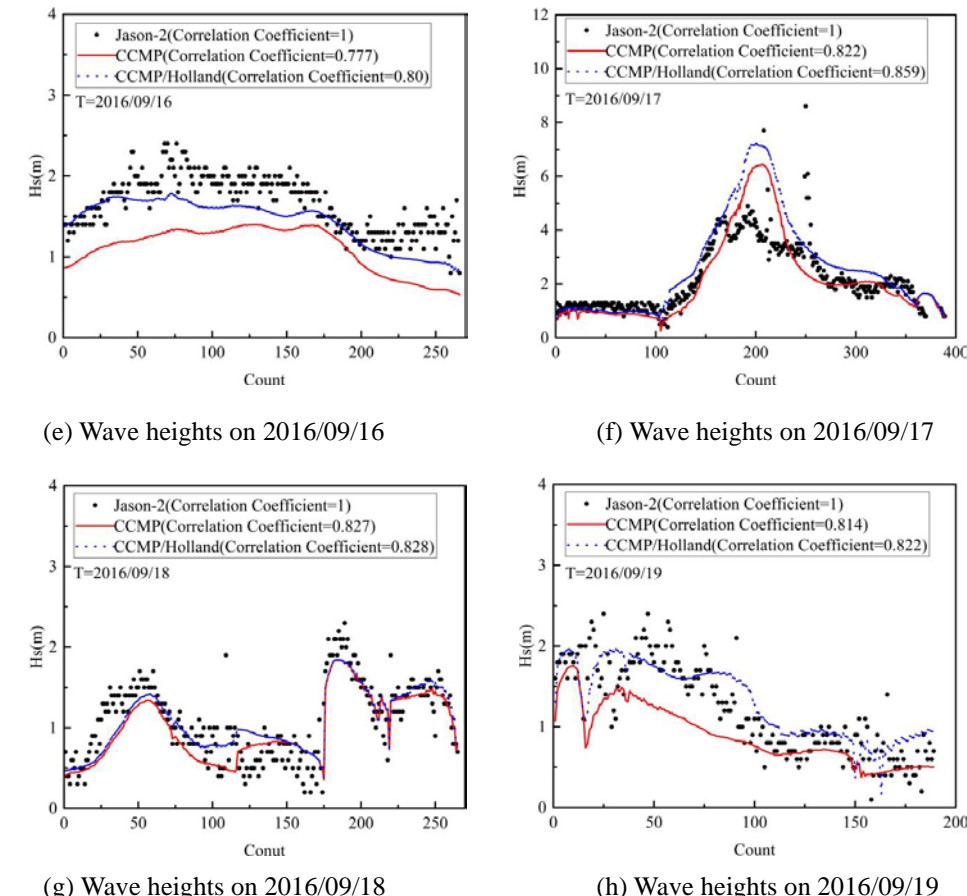


(e) Wave heights on 2016/09/16  (f) Wave heights on 2016/09/17

(g) Wave heights on 2016/09/18  (h) Wave heights on 2016/09/19

Fig. 11 Comparison of simulated wave heights driven by different wind fields and satellite observations

Table 3 Comparison of simulated wave heights and data of Jason-2

| Date | Deviation of wave height | CCMP | CCMP/Holland |
|---|---|---|---|
| 2016/09/12 | CC | 0.766 | 0.862 |
| | Bias (m) | -0.557 | -0.211 |
| | RMSE (m) | 0.596 | 0.309 |
| 2016/09/13 | CC | 0.961 | 0.839 |
| | Bias (m) | -0.746 | -0.073 |
| | RMSE (m) | 0.830 | 0.519 |
| 2016/09/14 | CC | 0.888 | 0.921 |
| | Bias (m) | -0.288 | 0.096 |
| | RMSE (m) | 0.519 | 0.402 |



|  |  |  |  |
|---|---|---|---|
|  | CC | 0.852 | 0.955 |
| 2016/09/15 | Bias (m) | -0.709 | 0.118 |
|  | RMSE (m) | 1.206 | 0.477 |
|  | CC | 0.777 | 0.80 |
| 2016/09/16 | Bias (m) | -0.557 | -0.226 |
|  | RMSE (m) | 0.596 | 0.309 |
|  | CC | 0.822 | 0.859 |
| 2016/09/17 | Bias (m) | -0.230 | 0.229 |
|  | RMSE (m) | 0.470 | 0.606 |
|  | CC | 0.827 | 0.828 |
| 2016/09/18 | Bias (m) | -0.081 | 0.030 |
|  | RMSE (m) | 0.273 | 0.215 |
|  | CC | 0.814 | 0.822 |
| 2016/09/19 | Bias (m) | -0.311 | 0.096 |
|  | RMSE (m) | 0.460 | 0.333 |

It can be seen from Figure 11 that the trend of the simulated wave heights driven by the CCMP wind field is consistent with

that for the Jason-2 data, and the correlation coefficients are more than 0.75; however, the simulated wave heights are slightly smaller than the observed data overall. The simulated wave heights driven by the blended wind field are much more consistent with the Jason-2 altimeter data than those driven by the CCMP, especially in the case of extreme waves. In general, the correlation coefficient of the blended wind field is higher than that of the CCMP. Table 3 shows that the Bias of CCMP ranges between -0.746 m and -0.081 m, which indicates that the simulated wave heights of CCMP are smaller than the actual values.

The bias of the blended wind field lies between -0.226 m and 0.03 m with a small average deviation and a significant decrease compared with the corresponding value for the CCMP. A comparison of the RMSEs of the two wind fields shows that the RMSE of the wave height driven by the blended wind field is, on average, reduced by 0.223 m compared to that for the CCMP. These results demonstrate that the blended wind field established in the present study can effectively improve the accuracy of simulation of waves during the three typhoons, and the blended wind field can more accurately simulate the wave field during

the three typhoons.

The comparisons of the simulated wave heights driven by the blended wind field and those driven by the CCMP from September 12 to 19, 2016 are shown in Figure 12. It can be seen from Figure 12 that the wave heights during the three typhoons in the north-eastern region of the South China Sea, the East China Sea and the Yellow Sea are relatively high, overall. Typhoon Meranti crossed the Bus Strait and landed in the Fujian Province, and it subsequently moved along the north of the Taiwan

Strait; in this course, the typhoon had a considerable impact on the cities along its tracks in China. The wind speed of Meranti was significantly high; such high wind speeds caused extreme waves near the Bashi Channel and north-eastern South China

 

Sea, and the maximum wave height exceeded 12 m. The wind speed of Meranti continued to decrease as it crossed the Bashi
Channel and Taiwan Strait because of the blocking effect of the two sides, and the wave heights further reduced with reduction
in the wind speed. Typhoon Rai 1615 originated in the southwest region of the South China Sea, with a low intensity and
relatively small impacts, and it mainly affecting the coasts in Hainan and Guangxi; the wave heights in these zones were
approximately 1–4 m. Typhoon Malekas followed typhoon Meranti closely. There was some distance between the two typhoon
centres, and no mutual rotation occurred; however, interactions between the two typhoons took place. Typhoon Malekas
primarily affected the eastern region of Taiwan, the East China Sea and the Yellow Sea, and it generated large waves with
maximum wave heights of approximately 12 m.

It was found that the simulated wave heights near the typhoon centre of the blended wind field were notably higher than those
near the typhoon centre pertaining to the CCMP wind field. The maximum wave height driven by the blended wind field was
more than 12 m, while that driven by CCMP was less than 8 m. From the Jason-2 satellite data, it can be seen that the simulated
wave height driven by the CCMP was relatively low, especially near the typhoon centre. From Figures 12 (c1) and 12 (c2), it
can be seen that in the CCMP wind field, typhoon Meranti 1614 arrived near the sea in Taiwan and subsequently moved
directly northward along the northeast region of Taiwan. The simulation results of the blended wind field indicated that Meranti
crossed the Bashi Channel after arriving in Taiwan and affected coastal zones, such as the Fujian, Guangdong and Zhejiang
provinces in China. The blended wind field was also consistent with the track of typhoon Meranti reported in existing literature
(https://baike.so.com/doc/6580510-24645572.html). It was thus noted that not only could the blended wind field effectively
improve the accuracy of typhoon wave simulation, especially for extreme waves, but it could also more accurately simulate
the moving track of typhoons.

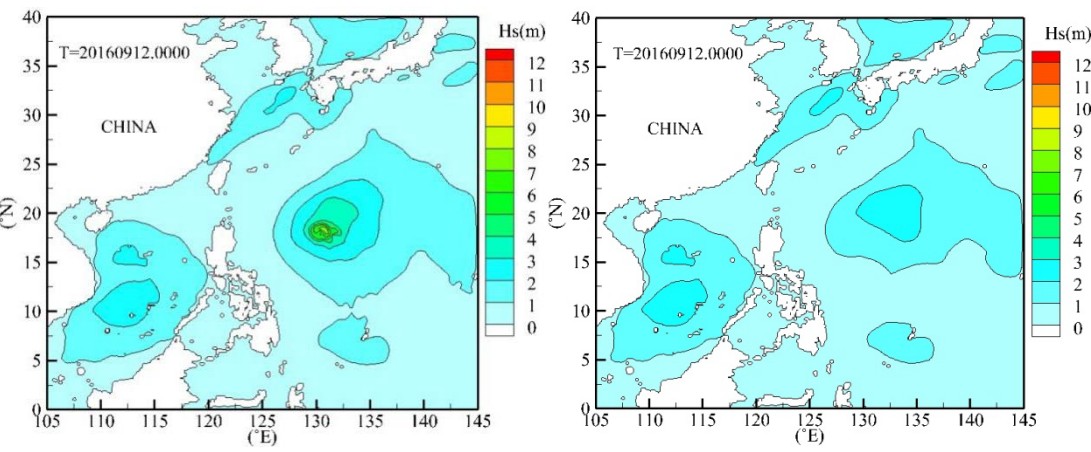

(a1) wave height on 20160912.00 (blended)        (a2) wave height on 20160912.009 (CCMP)





(b1) wave height on 20160913.00 (blended)     (b2) wave height on 20160913.00 (CCMP)

(c1) wave height on 20160914.00 (blended)     (c2) wave height on 20160914.00 (CCMP)

(d1) wave height on 20160915.00 (blended)     (d2) wave height on 20160915.00 (CCMP)





(e1) wave height on 20160916.00 (blended)     (e2) wave height on 20160916.00 (CCMP)

(f1) wave height on 20160917.00 (blended)     (f2) wave height on 20160917.00 (CCMP)

(g1) wave height on 20160918.00 (blended)     (g2) wave height on 20160918.00 (CCMP)





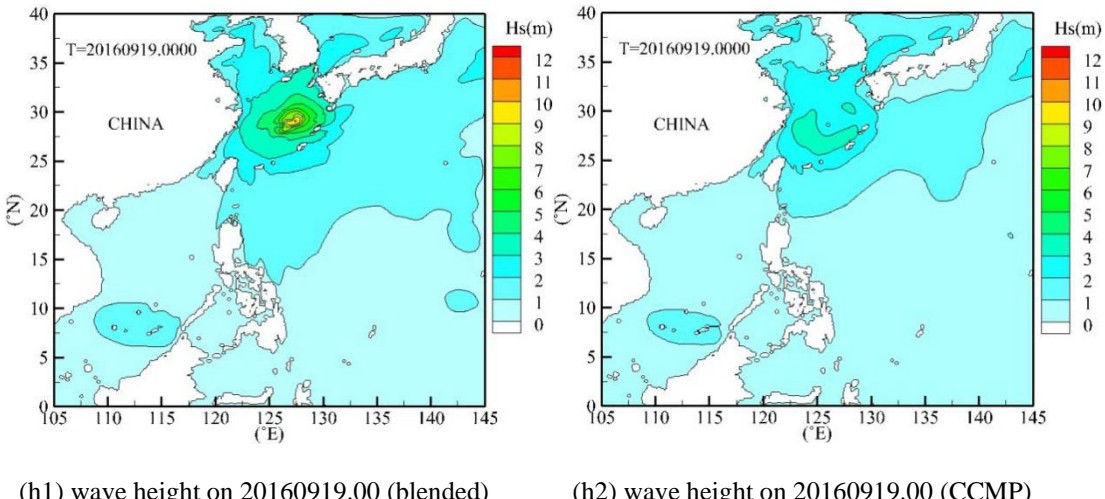

(h1) wave height on 20160919.00 (blended)   (h2) wave height on 20160919.00 (CCMP)

Fig. 12 comparison of wave field calculated by CCMP wind field and blended wind field

## 4. Conclusions

In this study, simulations of typhoon waves during three typhoons that occurred in China Sea, driven by different wind fields, were investigated. The typhoon waves driven by the ERA, CFSv2 and CCMP wind fields were simulated via SWAN considering three typhoons that occurred in the China Sea in 2015. Next, the maximum wind speeds near the typhoon centres during three typhoons that occurred in 2015 and 2016 were evaluated by comparing the values with the BEST TRACK data. Furthermore, the simulated wave heights driven by the CCMP and CCMP/Holland blended wind fields during three typhoons that occurred in 2015 and 2016 were analysed. The following conclusions could be obtained:

The typhoon waves in the China Sea during the three typhoons, driven by the ERA, CFSv2 and CCMP wind fields, could be simulated well. The simulated wave heights driven by the CCMP were more accurate for small waves. The simulated wave heights driven by the CFSv2 were more accurate for large waves, and the simulated wave heights driven by the ERA and CCMP were relatively smaller than the actual data. The maximum wind speeds near the typhoon centre of the three typhoons that occurred in 2015 and 2016 were compared with the BEST TRACK data, and it was found that the wind speeds of the ERA, CFSv2 and CCMP wind fields were relatively smaller than the corresponding BEST TRACK values. The Holland model was combined with the CCMP wind field, and the correction factor for the Holland model for the three typhoons Chan-hom, Linfa and Nangka that occurred in 2015 were 0.82, 0.78 and 0.78, respectively; the combined radius R was two times the maximum typhoon radius. The correction factors of typhoons Meranti and Malakas that occurred in 2016 were both 0.8, and the combination radius R was two times the maximum typhoon radius. To maintain the asymmetry of the original wind field, no correction was made for typhoon Rai because of its low intensity and short duration, and the maximum wind speed was consistent with the BEST TRACK values, overall.





Moreover, the blended wind field was noted to be superior to the CCMP wind field for the simulation of extremely large wave

heights. In particular, in the simulation of the typhoon waves caused by three typhoons in 2016, the correlation coefficient of the mixed wind field was higher than that of the CCMP wind field. The simulated wave heights driven by the CCMP were relatively small, and the Bias of the wave heights driven by the blended wind field was lower than that driven by the CCMP wind field; the RMSE of the blended wind field was reduced, on average, by 0.223 m compared to that of the CCMP. Furthermore, the wind speed pertaining to the CCMP wind field near the centre of the typhoon was relatively small, and the

position and moving track of the typhoon eye were deviated. The CCMP/Holland blended wind field could overcome these shortcomings and effectively improve the simulation accuracy of the wave field, especially for extreme waves.

### Acknowledgments

This work was supported by the National Natural Science Foundation of China (Grant Nos 51579036 and 51779039), and the Fundamental Research Funds for the Central Universities of China under contract No DUT19LAB13.

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
