# Peer review of "Numerical Investigation of Typhoon Waves Generated by Three Typhoons in the China Sea"

_Ocean Science, 2019_

## Referee Comment (RC1) · Anonymous Referee #1 · 31 Oct 2019

The paper "Numerical Investigation of Typhoon Waves Generated by Three Typhoons in the China Sea" touches an interesting topic such as wave generation under the simultaneous action of 3 typhoons. However, in its present form it is too descriptive and with limited critical assessment. Moreover, the presented analysis is only statistical with very limited mention to the underlying physics. Because of that I would recommend performing a more in depth physical analysis, combining physics with statistics to advance the state of the art on wave generation from winds with different directions acting on pre-existing swell also with different directions. That complexity, associated to the presence of 3 typhoons, would really advance the state of the art and provide for an interesting paper.

More specifically, I would suggest:

[Figure]

- Defining some of the concepts presented (for instance, a "complex mixed wave" on line 31) or the "Holland model" on line 50 (in this case defined much later in the paper)

- Emphasizing the applicability of such a complex wind and wave pattern form the China Sea to other seas in the planet.

- I would recommend reducing greatly the description of the SWAN model. However presenting in more depth the sink and source terms and the employed numerical discretization and the diffusivity that results would provide an interesting piece of research work. Particularly when assessing how it performs under various cyclones.

- The same regarding the selection for the wind energy input term (the same would apply to the sink-term) where the A and B coefficients have been selected from the state of the art without any mention to the presence of the cyclones.

- The performed comparison emphasizes correlation coefficients, bias, root mean square error, etc. A similar emphasis should be given to the underlying physics. Also, duly acknowledging the state of the art, where the fact that errors get larger when getting closer to the coast is already well known.

- On line 206 it is mentioned that the trend of the wave height is similar to that of the wind speed. Here no mention is made of the pre-existing swell from more distant cyclones and that would be an essential element for the analysis.

- In the comparison only wave height and wind speed are mentioned. Wave direction should here play a critical role. The same regarding wave period or wave age.

- When presenting the superposition formula (for equations 19 and following) the physics behind the proposed formulation and a comparison with other alternatives should be included.

- The same regarding the improvement of the blended wind field when compared to others (e.g. CCMP). The role of fit parameters should also be mentioned in here.

- When presenting the accuracy of the simulated wave field due attention should be paid to the underlying physics since otherwise the statement in line 311 and following that the simulated heights from the blended wind fields are much more consistent with Jason-2 than those from CCMP lacks meaning.

- The same regarding the conclusion (line 363) that the wave results driven by CCMP are more accurate for small waves and the opposite for CFSv2.

In summary the paper should emphasize more the physics behind the analysis and be more critical with respect to the obtained results. With that I think it could provide an interesting contribution to the state of the art.

---

## Referee Comment (RC2) · Anonymous Referee #2 · 7 Nov 2019

The manuscript describes a hindcast of typhoon generated waves in the China Sea using a third-generation wave prediction model. Attention is given to the choice of the wind data source and comparisons are made of hindcast data against nearshore buoy data and satellite data. My general comment is that the manuscript is too descriptive without having a proper research question. Expected differences between different model setups are noted and trivial conclusions are drawn. Reading between the lines it appears that the simultaneous occurrence of 3 typhoons is the challenging part of this work. This could be the starting point of a more in-depth analysis addressing the challenges of performing a hindcast for such a situation and giving attention to the blending of wind fields, choosing the proper wave physics and extending the analysis also to other wave parameters like mean/peak wave direction, and wave period measures.

The chosen frequency range is 0.04 Hz – 1 Hz. I wonder whether the low frequency limit is properly chosen as this relates to waves with a period of 25 s. It could be possible that under typhoon conditions longer wave periods occur. This issue should be discussed in combination with wave period information (e.g. Tm-10, Tm01) As 3 typhoons interact with each other, the resulting wave field is expected contain various swell fields and an analysis of the 1D- and 2D frequency spectra could provide relevant insights.

More detailed comments are as follows:

The title contains twice the word typhoon and it misses the word simultaneously. Please reconsider the title.

The numbers 1509, 1510 and 1511 in line 7 should be introduced. I assume it is some local numbering of typhoons?

Line 29: it is not clear what kind of evolution laws are meant in complex situations. I do not think parametric growth laws apply here.

The description of the wave model should be revised. Eq. (3) is not needed as no currents are part of this study. Part of the model settings are default, but of more interest are the physical settings relevant for typhoon modelling like the choice of wind forcing, wind drag relation and whitecapping dissipation. Further, as the model is (presumably) run in non-stationary mode, details should also be given about the time stepping, convergence settings (applicable to SWAN), and the spatial resolution. Evidently, no tuning of the wave model is done. That could be a good starting point, but in view of the noted discrepancies, tuning should be mentioned, if possible, as a way to improve modelling results. Evidently some nesting is applied, but no details are given of the extent of such nest areas. This should be made more clear in Figure 2.

Figure 3 should be supplemented with a table providing the positions and depths of the various wave buoys.

In line 168 and elsewhere it is not explicitly stated that the wave height is the significant wave height. Moreover, providing only information on this wave parameter is too limited. Especially directional and period information supplement the analysis.

Line 175 mentions diffraction and fragmentation. I do not think diffraction will play a role on the scale of the China Sea, and it is unclear what is meant with fragmentation.

Line 186 contains a negative bias where 2 lines earlier the phrase absolute values was used. This is confusing.

As a general comment here and elsewhere I notice that BIAS and RMSE are given in 3 decimals. I do not believe this to be realistic. The number of decimals should reflect the accuracy of numbers.

The legends of many figures are incomplete. Please make these more descriptive.

The results in Table 2 show differences in performance. This can be expected, but the consequences of these differences, related to the still unknown research question, are unclear. The number of hindcasts is too limited to draw sound conclusions about the best wind data source.

Figure 5 shows many discrepancies between measured and modelled winds. Do the authors consider this a sufficiently good basis for performing the hindcast?

The superposition of the various wind fields in the Holland model is still unclear. Of particular interest is whether the combined wind field (in combination of the superimposed pressure fields) is physically possible. Hinting at Wang (2017) is not helpful as this is in Chinese language and not accessible for the interested reader. This manuscript could be a place to make an accessible presentation of this topic. See also my previous comment on the challenges.

Line 254 mentions some numbers. It is unclear in what way these numbers were obtained.

What is the basis for the statement in line 264 that the blended wind is more accurate?

The lines 268 and 275 (as well as at other places) use rather qualitative terms 'are similar' or 'more reasonable'. The criteria for making such statements are missing. And using more quantitative measures would improve the content.

Figure 8 shows the paths of the 3 typhoon that more or less seem to occur simultaneously. It would be good to plot a few parameters of each typhoon in one plot with time along the horizontal axis. Then more attention can be given to the mutual timing of each typhoon.

Same comment for Figure 9.

What is meant with 'mutual rotation' in line 332?

The abundance of subplots in Figure 12 is not needed. These figures show differences, but what is exactly the message. Why not pick a few time instances to answer this and discuss those in detail. In addition, plotting vectors with the mean wave direction may reveal some of the complexity of such a hindcast.

---

## Referee Comment (RC3) · Anonymous Referee #3 · 14 Nov 2019

A numerical simulation of combined waves generated by a series of three typhoons in the China Sea in 2015 was described by the article. A wave model based on a third generation wind-wave model was used where the driving wind field was derived from three types of data sets. The simulation of multiple typhoon waves involves, admittedly, complicated computations and difficult to attain high level of accuracy. Furthermore, it has not been rigorously investigated and validated before. Thus this article presented a methodology that would be of interest to those investigating the impact of typhoons acting in tandem or simultaneously. To what extent this has been achieved needs to be thoroughly discussed including the fundamental aspects of the physical processes that occur. The paper should cut to the chase and focus on the novelty of the investigation, which is the simulation of waves resulting from three simultaneous typhoons.

[Figure]

The following comments and suggestions are given:

Description of the model set-up should be complete with details of the three typhoons and how these were implemented in the model. Do the three typhoons occur exactly simultaneously during the duration of the simulation period from 1st to 18th July 2015. Describe the path-lines of the three typhoons and the significance of the location of the four buoys. Same comment applies to the typhoons shown in Figure 8.

Referring to Figure 4 that compare wave height variation at the buoy stations, it was observed that the time periods (T Day/Hour) are different for each stations. Explain this.

It was also noted that buoy #4 is the closest to Linfa (1510) path. Would this has any bearing to the results at buoy #4? Considering it proximity to the path of Linfa and comparatively sheltered by islands, would a single typhoon simulation (of Linfa only) produce a better correlation? The statements between lines 171 to 178 did not touch on this point.

LINE 225 – how was CCMP justified as giving the best performance?

LINE 254 – explain how the K-values were determined for the three typhoons.

Figure 7 – which line is before and which line is after correction of wind?

Figure 8 - label should be "in 2016"

LINE 321 – comparison between blended/CCMP were made in Figure 12. Some differences were noted. How do they compare with observed data?

---

## Referee Comment (RC4) · Anonymous Referee #4 · 18 Nov 2019

This paper summarizes the results of the wave simulations for three almost simultaneous typhoons in the China Sea by comparing and contrasting the results for three different sources for the input wind field. Simulation results were compared with coastal buoy observations and satellite data of wave height and the range of applicability for simulations from each wind field was determined. Also, the maximum footprints of wind field were compared with the Best-Track data and a combination of Typhoon wind field from Holland methods with one of the wind fields was used to modify the underestimation of wind speed close to the center of typhoons as observed for the three used wind field. Although some interesting results were presented regarding examining the accuracy of wind fields and the blending approach with the Holland method, the paper obviously lacks a coherent research outline and a scientific story and with the present

form cannot be recommended for publication in the Ocean Science. The authors need to apply major modifications to change the paper to the appropriate form for publishing in this journal. Followings are more details about the issues with the paper in the present form that need to be addressed by the authors:

Overall comments The major problem of the manuscript is that no appropriate research question considering the background of typhoon wave modeling in the China Sea was asked and the innovation and scientific merit of the research is not clearly specified. The authors claim that that the difference between their research and previous studies is that in the new study three simultaneous typhoons were considered in the comparison of a single typhoon in other studies. However, the authors failed to elaborate on the differences in simulation for single and three-typhoon waves. In fact, there shouldn't be any difference in the simulation of waves generated by a single typhoon with three typhoons. The simulation approach is the same and the same sources for wind fields are used. The main complexity would be the generated wave field and the interaction of different wave systems generated by different typhoons that the authors again failed to investigate.

A scientific paper needs a proper discussion section (could be embedded in other sections) to apply the results in the context of a broader scientific contribution. Simulation of a wave model with different wind fields and evaluating the results to find the validity of each wind field is an interesting observation. However, this is only the result part of the paper and is not actually a scientific contribution. The authors only compared wind and wave results from different sources and did not include any further scientific analysis. The discussion could be on the spectral pattern of typhoon-generated waves during the effect of three typhoons (as an example please see Allahdadi, M. N., Chaichitehrani, N., Allahyar, M., and McGee,L.: Wave Spectral Patterns during a Historical Cyclone: A Numerical Model for Cyclone Gonu in the Northern Oman Sea, Open Journal of Fluid Dynamics, 7, 131, https://doi.org/10.4236/ojfd.2017.72009, 2017) or interaction of swell waves generated by each hurricane and their effect on the wind input/whitecapping dissipation in the model ( example: van der Westhuysen, A. J., Zijlema, M., and Battjes, J. A.:Nonlinear saturation-based whitecapping dissipation in SWANfor deep and shallow water, Coast. Eng., 54, 151–170) or other modeling related issues that can used the present modeling results based on the authors' selection.

If the authors claim that their innovation is evaluating different wind fields used in wave modeling, still they need some level of discussions about the wind fields, their complexity during the three simultaneous typhoons, and the technical reasons that the wind data do not present a high accuracy wind field during the typhoon events. One thing that can helpful for the discussion is comparing the wind timseries from each typhoon with timesries of observed winds at the buoys like the buoys used in this paper or even NDBC buoys if the data are available in the study area by establishing scatter plots and supporting them with the model performance statistics.

The introduction part of the paper is not coherent and was not appropriately outlined. This just a list of simulation works or verifying wind data that are not necessarily related to each other and are not narrowed down to get to the main research question. Please consider rearranging the introduction based on really relevant papers that could appropriately introduce the main research question of the paper.

Some important details about model setup and configuration are missing.

The text is not clear at many parts. Some sentences are incomplete, some technical terms were selected wrongly, and many grammatical and verbal errors are found. Please fix these problems in the modified version of the paper (more details will be presented below).

Details Ln 9, 13,40: Different words were used for representing ERA wind field(ERA, ERA-I, and ERA-Interim) please be consistent and use only one of them throughout the paper. Ln 20, 21: "root mean square of the blended wind field was 0.223 m lower than that of the CCMP" : do you mean the root mean square of the simulated wind field by each of these wind fields? Please modify the sentence Ln 25: you need to

elaborate more on "complicated" and present more information about how the three-typhoon wave field and why is it complicated. Ln 26: Generation of Typhoon although it has some pre-conditions that make it more likely in the summer, is basically a random process. Why three-typhoon events happen almost every year in the China Sea? What conditions cause them to happen together? You need to add some clarifications here Ln 29 and 59,60: What do you mean by the "evolution laws" ? Do you mean the "wave evolution process"? Ln 35: what do you mean by: plane distribution"? Shouldn't it be:" spatial variations"? Ln 40: what do you mean by" the same set of altimeter and buoy data"? the same as what data? Ln 44, 56: Please be consistent with using WW3 or WWIII. Also, for the first use of this acronym in the text use the full word: WAVEWATCHIII. Ln 47-50: If the study of Shao et al(2018) is a modeling study, please mention it and present more details including the numerical model; that was used in this study. Ln 60-62: The authors did not present any details about the kind of investigation that should be implemented in the case of three-typhoon events. What has been done specifically in this research to fulfill a part of this investigation? Entire manuscript: too much use of the word "pertaining". In some cases, even the selection of this word is not appropriate. Ln 80: what do you mean : "wave increase and decrease"? do you mean" wave generation and dissipation"? Ln 80: please change: control equation" to "governing equation" Ln 92: k is not the unit vector. It is actually the wave number vector. The magnitude of the vector is k=$2\pi$/L in which L is the wave length. Ln 100, 101: Sds,mud is the dissipation term to include the effect of wave-mud interaction. Sediment dissipation for sand sediments is included in ds,b

Section 2.1. The author need to include more details about the SWAN model type and the setup used for the simulation. The followings should be addressed in the test: What version of SWAN is used? Is it based on a structured or unstructured model? What wind input and whitecapping approaches were used in simulations and why? What kind of boundary conditions were used and from which source? If no boundary condition is used present proper reason(s) or references(s).

Ln 117: The spatial resolution of the ETOP1 bathymetry data is about 1600-1800meters depending on the latitude which makes it suitable only for modeling pur-posed in the offshore areas where the depth gradients are not important. Since in the present study, several coastal buoys is used for model verification, higher accuracy bathymetry data may be required. Have you checked for such data?

Ln 119-132: please add a table to the manuscript summarizing all three wind fields used in this study with information about: spatial resolution, temporal resolution, and the association that handles the data.

Section 2.3 title: please change the title to:" Modeling area and setup"

Ln 142: change "range of calculation: to "modeling area"

Ln 144: change " topography" to "bathymetry"

Section 2.3 : Please either add a table showing buoy characteristic (coordinates and water depth) Figure 1 or 2: mark the approximate location of the South China Sea, Yellow Sea, and East China Sea on one of these figures. Figure 1: for each typhoon of 2015, mention the start and end date on the figure or in the manuscript. Ln 150-151: how did you choose the spectral characteristics of the model? Please elaborate more ad add appropriate references. Ln 150: Regarding the extensive modeling area and the relatively large directional resolution of the model(10 degrees), the simulation results may be affected by the garden sprinkling effect(GSE). Have you done any sensitivity study on the directional resolution to check the model results for this effect? Ln 151-152: what about whitecapping dissipation? Did you include this term? Ln 153: what are the value of timestep and the number of computational iterations used in the SWAN model? Ln 155: Same as the main model, for the nested model some details about the model setup should be mentioned. General comments about the caption of figures with multiple panels: all the caption should be written below the main figure and the panels should be mentioned with a,b,c,. within the same caption not separately. While the panels are marked with a,b,c, . . . Equations (13) and (14): parameter n was

not defined. Ln 175: what is "fragmentation"? Ln 166-167: this sentence is not clear. How the orthogonal representation of the coastline caused inaccuracy at the coastal buoy? Was a curvilinear SWAN model used in the simulation? It has not been mentioned anywhere in the manuscript. Ln 171-178: The reasons that are presented for lower accuracy of simulations at buys in the South China Sea compared to the Yellow sea are very general and cannot be verified without knowing the water depth at each buoy and their distance from the shoreline. We know that simulation results at the coastal buoys are not generally as accurate of offshore buoys. Aside from refraction and other shallow water effects, two more important effects degrade modeling results at coastal buoys: interpolation of wind data with land and the "Slanting fetch" effect (please see: Allahdadi, M.N., He, R., Neary, V.S., 2019. Predicting ocean waves along the US east coast during energetic winter storms: sensitivity to whitecapping parameterizations. Ocean Science 15, 691–715. https://doi.org/10.5194/os-15-691-2019). The authors need to appropriately discuss and mention these reasons and present related references. Ln 180-183: These sentences are not clear. Please elaborate more. Ln 196: "trajectory point" can be changed to "observation point"

Ln 207-209: These sentences need to be modified for clarity Ln 230: What is "B(1980)"? Ln 253-245: describe how K values for different typhoons were calculated? If there is any formulation please include it. Ln 60: add snapshots of the blended wind field and the unmodified wind field to show wind speed contours and wind vectors for two cases Ln 300: change "trajectory points" to "observation tracks" Table 3: combine all the results for different days into one scatter plot (satellite-model) with statistical values instead of the table. The table is not very useful when you separate different days. Ln 313: do you mean correlation coefficient of simulated waves? Ln 345: what do you mean by "moving track of typhoon"?

I hope these comments are helpful for improving the revised paper.